# OpenReview forum: "Adaptive Activation Steering: A Tuning-Free LLM Truthfulness Improvement Method for Diverse Hallucinations Categories"
_ACM.org/TheWebConf/2025/Conference — WWW 2025 Poster_

### Official Review · Reviewer_81KL · 2024-11-21

**Novelty:** 4
**Technical Quality:** 5

**Review:**

This paper presents a method called Adaptive Activation Steering (ACT), which aims to address the gap between "knowing" and "expressing" in large language models (LLMs) and improve their realism. The method calculates steering vectors and utilizes diverse steering vectors and adaptive steering intensity control during inference to steer the model activations towards more realistic directions. The experiments were conducted on the TruthfulQA benchmark and involved multiple models, showing that ACT improved the realism of the models. Additionally, the effectiveness was further validated through manual evaluation. The paper also explores the generalization ability of ACT on datasets beyond TruthfulQA, finding that ACT also improved on other dataset.
Finally paper points out the limitations of ACT. Generalities in multi-round conversation scenarios are not explored.The tradeoff between usefulness and authenticity is also not explored truthfulness.

### strengths
1. The paper  describing the problem and previous studies very clearly.
2. The paper proposes an adaptive and diverse steering vector to better guide the output of large models towards true answers.

### weakness
1. In the experiment, there was no comparison between using only FSP and using only ACT in full data setting.It is impossible to determine which method in the paper performs better than FSP.
2. Using ACT on the basis of FSP does not prove that these two methods are orthogonal.If the two methods are not orthogonal, there is no guarantee that using both methods simultaneously will result in the best possible outcome.
3. Without creativity-relevant tests on the modified output, it is impossible to determine whether the use of ACT has an impact on the creativity of the model.
4. The definitions of the probes in Equation 2 and Section 3.2 seem to conflict; Equation 2 uses $x_l$ while Section 3.2 uses $a_l^h$.

**Questions:**

1. Is it reasonable to select the top K probes based on their accuracy on the validation set? If there is a probe with low accuracy but with a larger value, would it be better to give it some weight?
2. Relying solely on the output of the last token may lead to difficulties in addressing issues with longer text lengths. Would it be better to set a length and uses multiple steerings?

**Reviewer Confidence:**

2: The reviewer is willing to defend the evaluation, but it is likely that the reviewer did not understand parts of the paper

**Scope:**

3: The work is somewhat relevant to the Web and to the track, and is of narrow interest to a sub-community

---

### Official Review · Reviewer_5ajs · 2024-11-28

**Novelty:** 4
**Technical Quality:** 4

**Review:**

### Summary
This paper introduces Adaptive Activation Steering (ACT), a novel method designed to enhance the truthfulness of LLMs without the need for additional tuning. ACT adaptively adjusts the activation directions during inference to steer LLMs towards more truthful outputs, addressing a variety of hallucination categories. Experiments shows ACT can improve truthfulness of generation across several LLMs and is shown to be scalable across different model sizes.

### Strength
1. This paper propose a new method ACT to improving LLM truthfulness without requiring model retraining or fine-tuning.
2. Experiments shows that ACT can effectively improve trustfulness across different datasets.

### Weakness
1. Despite the authors' explaination of the work's relevance to the Web in the Appendix A, the connection remains tenuous. The paper's focus on improving LLM truthfulness, while important, does not substantially address Web-specific challenges or directly contribute to the advancement of Web technologies or infrastructures. This limited relevance may reduce the impact of the work within the context of a Web-focused conference. Therefore, I suggest that the paper might be more appropriately suited for NLP conferences like ACL.

2. The work primarily experiments only on the TruthfulQA dataset. It is recommended that additional experiments be conducted on a wider range of datasets, such as RAGTruth, HotpotQA, and Halueval, to better assess the generalizability and robustness of the proposed ACT.

3. The paper could benefit from a clearer rationale behind the chosen evaluation metrics and including citations.

4. Experiments, aside from Table 1, lack comparisons with baselines such as ITI, which is essential for evaluating the method's performance.

**Questions:**

Please see the weakness.

**Reviewer Confidence:**

3: The reviewer is confident but not certain that the evaluation is correct

**Scope:**

2: The connection to the Web is incidental, e.g., use of Web data or API

---

### Official Review · Reviewer_VGKP · 2024-12-01

**Novelty:** 6
**Technical Quality:** 6

**Review:**

This paper presents an interesting and import problem in Large Language Models: how to improve the truthfulness and reduce the hallucinations in LLM. The authors provide a tuning free method to achieve this goal. They consider the diverse categories of hallucinations and utilize the diverse truthfulness-related steering vectors to improve the truthfulness. Considering the change of the training process, they derive a dynamic adaptive approach. The experiments carried by the authors demonstrate the improvement of this method.

Strengths:
1.	This paper considers the hallucination problem in LLM, which is significant in the related field.
2.	It provides a tuning free and adaptive approach, and at the same time considers diverse hallucination categories, which can be practical in the application.
3.	The experimental results show the significant improvement on various models and also verify the scalability of the proposed method.

**Questions:**

1.	To generate different probs for diverse hallucination categories, the K-means clustering method is used. It is noticeable that the hyperparameter cluster number C can affect the performance. In real setting, how to obtain this cluster number should be discussed.
2.	In the experiments, some tuning-based methods are proposed as baselines to the tuning-free method, the difference of the efficiency among them should be discussed.

**Reviewer Confidence:**

3: The reviewer is confident but not certain that the evaluation is correct

**Scope:**

4: The work is relevant to the Web and to the track, and is of broad interest to the community

---

### Official Review · Reviewer_B8jV · 2024-12-02

**Novelty:** 5
**Technical Quality:** 4

**Review:**

Paper summary: This paper introduces ACT, a tuning-free method that enhances the truthfulness of large language models (LLMs) during inference. ACT uses truthfulness-related steering vectors to adjust activations dynamically, without fine-tuning, and adapts steering intensity based on content truthfulness. Extensive evaluations on the TruthfulQA benchmark demonstrate significant improvements.

Strength:
1.	The authors propose a simple yet effective strategy to attend to truthfulness in LLM output. Extensive evaluations demonstrate its superiority.
2.	The authors deploy a great practice to have human evaluation for a comprehensive comparison. This part makes the results stronger.

Weakness:
1.	The gain by ACT is counter-intuitive to me when we compare LLaMA2 and LLaMA3. Particularly, powerful models have greater improvement when equipped with ACT.  The powerful model should be stronger in most aspects and may not need ACT to further adjust truthfulness. The improvement should be minimal to some degree.
2.	One clarification question: Since LLaMA 3.1 came out in July 2023, did authors have experiments on that part?
3.	The presentation needs improvement, including 1) broken sentences (e.g., “express fully” and “despite their fluency” on Page 1); 2) grammar errors (e.g., “adjust” -> “adjusts” on Page 1)
4.	The authors propose an adaptive steering strategy. They can illustrate more on why this adaptation works better. Otherwise, the proposed method is not technically strong with a novel contribution.
5.	The authors mention and deploy the k means method in Section 3.3. How do they choose k?
6.	More details should be given on human evaluation parts. How many experts rate one example? How do authors handle the bias among experts?
7.	Even if the author claims the generalization of the proposed ACT, it may be a good practice to test it on another dataset in another domain.

**Questions:**

Please see the weakness part.

**Reviewer Confidence:**

3: The reviewer is confident but not certain that the evaluation is correct

**Scope:**

3: The work is somewhat relevant to the Web and to the track, and is of narrow interest to a sub-community

---

### Official Review · Reviewer_yU9r · 2024-12-02

**Novelty:** 6
**Technical Quality:** 6

**Review:**

I would like to thank the authors for their work on an important topic such as reducing the hallucination in LLM responses.

**Summary of the work**
This work presents a framework names Adaptive Activation Steering (ACT) which is a tuning-free method that adaptively shifts LLM’s activations in the "truthful" direction during inference. ACT differs from the past frameworks as it takes in account of the diversity in steering vectors and their intensity. Authors conduct the evaluation on this framework on the TrurthfulQA dataset in multiple settings (open-ended generation and multiple choice QA). Finally, the work also presents results from the human evaluation, generalizability and scalability evaluations.

**Strengths of the work**
- The work is novel especially the idea of accounting for diverse set of steering vectors and also their intensity based on the Truthfulness of the already generated answers is intuitive and well-executed.
- Authors have conducted a comprehensive set of evaluations that goes beyond the standard evaluation of the TruthfulQA dataset. I appreciate the authors for taking an extra-step of conducting additional evaluations especially on diverse set of hallucinations computational efficiency and scalability.
- The writing of this work is clear given that it a dense paper with a large number of notations and numbers.

**Weakness of the work**
- While the work presents a comprehensive set of evaluations, I am a little concerned with the evaluation metrics. Although BLEURT is a popular approach, there are newer approaches that are being used for assessing alignment between a pair of texts (such as AlignScore [1]). Similarly, using GPT-3 as a judge for GPT based evaluation seems a little outdated, there are better models (GPT-4 etc) and better approaches (Prometheus 2 [2]) that could have been used for a more robust assessment of the performance of the models.

- Some details related to the human evaluation are missing. Specifically, it is unclear whether IRB approval was obtained for this evaluation. Additionally, the paper does not provide sufficient information about the experts involved, such as their fields of expertise, their recruitment process, and their qualifications in their domain. Given the diverse domain range (Figure 4), it would be essential to understand if the human evaluators were equipped with sufficient knowledge.

- Authors mention a 2% overhead using ACT. However, based on my calculations, this translates to a latency increase of
~3.8 milliseconds per query [calculation: (18.53-18.16)*60/5882]. This seems significant, especially given the real-time and scalability requirements of web platforms. Hence, I am doubtful about the applicability of this approach which is crucial for this work to be related to Web Conference.


**References**

[1] Yuheng Zha, Yichi Yang, Ruichen Li, and Zhiting Hu. 2023. AlignScore: Evaluating Factual Consistency with A Unified Alignment Function. In Proceedings of the 61st Annual Meeting of the Association for Computational Linguistics (Volume 1: Long Papers), pages 11328–11348, Toronto, Canada. Association for Computational Linguistics.

[2] Kim, Seungone, Juyoung Suk, Shayne Longpre, Bill Yuchen Lin, Jamin Shin, Sean Welleck, Graham Neubig, Moontae Lee, Kyungjae Lee, and Minjoon Seo. "Prometheus 2: An open source language model specialized in evaluating other language models." arXiv preprint arXiv:2405.01535 (2024).

**Questions:**

I have the following questions/requests:

- I am curious whether the authors could conduct the evaluation using more recent and robust metrics. Specifically, for GPT-based judge metrics, it would be advisable to utilize newer and more capable models (such as GPT-4). Less advanced models are prone to higher rates of hallucination, which may affect the reliability of the evaluation (which is my concern).

- I would request the authors to provide more details regarding the human evaluation round (see the weakness point 2).

**Reviewer Confidence:**

3: The reviewer is confident but not certain that the evaluation is correct

**Scope:**

3: The work is somewhat relevant to the Web and to the track, and is of narrow interest to a sub-community